# DEFECTIVE ENDOSPERM-D1 (*Dee-D1*) is crucial for endosperm development in hexaploid wheat

Natalia Tikhenko[1,2], Ahmad M. Alqudah [1,3 ✉], Lioudmilla Borisjuk[1], Stefan Ortleb [1], Twan Rutten[1], DanDan Wu[1,4], Manuela Nagel [1], Axel Himmelbach[1], Martin Mascher[1], Marion S. Röder[1], Martin W. Ganal[5], Stefanie Sehmisch[1], Andreas Houben [1] & Andreas Börner[1,3 ✉]

Hexaploid wheat (*Triticum aestivum* L.) is a natural allopolyploid and provides a usable model system to better understand the genetic mechanisms that underlie allopolyploid speciation through the hybrid genome doubling. Here we aimed to identify the contribution of chromosome 1D in the development and evolution of hexaploid wheat. We identified and mapped a novel *DEFECTIVE ENDOSPERM–D1* (*Dee-D1*) locus on 1DL that is involved in the genetic control of endosperm development. The absence of *Dee-D1* leads to non-viable grains in distant crosses and alters grain shape, which negatively affects grain number and thousand-grain weight. *Dee-D1* can be classified as speciation locus with a positive effect on the function of genes which are involved in endosperm development in hybrid genomes. The presence of *Dee-D1* is necessary for the normal development of endosperm, and thus play an important role in the evolution and improvement of grain yield in hexaploid wheat.

[1] Leibniz Inst Plant Genet & Crop Plant Res (IPK), OT Gatersleben, Corrensstr 3, D-06466 Seeland, Germany. [2] Vavilov Institute of General Genetics, Russian Academy of Sciences, Gubkina 3, 119991 Moscow, Russia. [3] Institute of Agricultural and Nutritional Sciences, Martin Luther University Halle- Wittenberg, Betty-Heimann-Straße 3, Halle (Saale) 06120, Germany. [4] Triticeae Research Institute, Sichuan Agricultural University, Chengdu, China. [5] TraitGenetics GmbH, 06466 Seeland OT, Gatersleben, Germany. ✉email: alqudah@ipk-gatersleben.de; boerner@ipk-gatersleben.de

Wheat is the second most important crop in the world, providing more than 20% of the world-wide calorie intake[1]. In 2017 world wheat production reached 759.4 million tons[2] mainly consisting of hexaploid wheat (95%) with minor contributions of durum wheat (4%) and emmer wheat (1%)[3]. The increment in ploidy level of wheat improves features such as productivity and adaptability to various growing conditions[4]. Hexaploid wheat (*T. aestivum* L.) is the product of the natural hybridization of three different diploid species: (i) an unknown *Aegilops* species (genome BB) closely related to *Ae. speltoides*, (ii) *T. urartu* (genome AA) and (iii) *Ae. tauschii* (genome DD) followed by genome doubling[5]. The first hybridization event between the B genome donor and *T. urartu* leading to the allotetraploid wheat *T. turgidum* subsp. *dicoccoides* (wild emmer, genome structure BBAA) occurred approximately 0.8 million years ago[5]. The second event which took place about 8000 years ago led to the formation of the hexaploid wheat *T. aestivum* (BBAADD)[6,7]. In such polyploids, gene duplication alters the transcriptional landscape by providing additional flexibility to adapt and evolve new patterns of gene expression for homoeologous gene copies[4]. In line with the model that the three homeologous genomes contribute equally to overall gene expression[8], it is suggested that the addition of the D genome to the A and B genomes led to a change in subgenome dominance[9]. Nevertheless, the true relationship between the individual genomes is still not well understood. Nullisomic-tetrasomic and deletion lines are often used to identify the contribution of individual chromosomes and loci towards features that determine the productivity and adaptability of wheat[10].

In modern breeding programs wild and cultivated relatives of wheat serve as a source of useful alleles for wheat improvement. In particular, rye (*Secale cereale* L.) provides vast genetic variation for commercially important traits such as stress tolerance, biomass, yield, and photosynthetic potential[8,11]. The success of targeted distant hybridization is primarily determined by the ability to overcome reproductive barriers. Since crosses between hexaploid wheat with rye are more effective than those between tetraploid wheat with rye it was assumed that the presence of the D genome is a prerequisite for the development of endosperm, ensuring the viability of hybrid kernels[12].

In our previous studies we have shown that, in crosses between hexaploid wheat and rye, the wheat chromosome 1D has a strong influence on embryo lethality and endosperm abortion during hybrid seed development. Since tetrasomy for chromosome 1B partially compensated for the absence of 1D, chromosome 1B may carry homoeoallele(s) of this 1D locus (or loci)[13].

In the present study, we further elucidated the role of chromosome 1D in the genetic control of wheat and wheat-rye hybrid endosperm formation and localized this unknown locus by physical mapping. Finally, we discuss the contribution of chromosome 1D in the shaping of spike traits and grain yield, as well as its role in the domestication and evolution of hexaploid wheat.

## Results

### Physical mapping of the locus (loci) underlying seed set and other traits in hexaploid wheat.
To evaluate the relevance of wheat chromosome 1D for seed development in wheat- rye hybrids, we carried out a physical mapping of the responsible locus (loci). The analyzed characteristics are listed in Table 1.

Each line was crossed with inbred marker line L6 or commercial rye populations to examine the presence of postzygotic barriers between hexaploid wheat and rye. Viable hybrid kernels were only produced in crosses involving the following lines: Chinese Spring (CS), N1D/T1B, Dt1DL, 1DL-8, and five deletion lines for the short arm of chromosome 1DS. Non-viable seeds were obtained in crosses with N1D/T1A, Dt1DS, 1DL-4, 1DL-1, 1DL-3, and 1DL-6 lines (Supplementary Fig. 1).

The lines that formed non-viable hybrid seeds (when crossed with rye) differed significantly in spike morphology and productivity from those lines that did produce viable ones, viz. CS, N1D/T1B, Dt1DL, five deletion 1DS lines, and the 1DL-8 deletion line (Table 1; Fig. 1). ANOVA of the main spike traits (Table 1) was evident for the existence of a specific locus (loci) located on the long arm of chromosome 1D in position low of − 0.29 breakpoint interval which is essential for seed development in both self-pollinated as well as in rye pollinated wheat. This locus is absent in the N1D/T1A and in deletion lines 1DL-4, 1DL-1, 1DL-3, and 1DL-6. All these lines have an extremely low

**Table 1 Characteristics of main spike (MS) of hexaploid wheat CS, N1D/T1A, N1D/T1B, Dt1DS, Dt1DL, and CS deletion lines of chromosome 1D (field experiment, 2015).**

| Line | Spike length (cm) | Number of spikelets | Number of seeds | TGW (g) | Seed set (%) | Seed viability[a] | Breakpoint interval[b] |
|---|---|---|---|---|---|---|---|
| CS | 8.2 | 21.4 | 63.8 | 35.3 | 87.3 | + | − |
| N1D/T1A | 8.8 | 23.4** | 25.2** | 14.5** | 43.8** | - | − |
| N1D/T1B | 8.4 | 21.4 | 52.2 | 29.1 | 78.9 | + | − |
| Dt1DS | 10.4** | 21.6 | 18.7** | 21.0** | 41.6** | − | Cen -1.0↑ |
| 1DS-5 | 8.3 | 19.8 | 56.2 | 33.8 | 82.7 | + | + 0.70 |
| 1DS-4 | 9.0 | 21.0 | 55.4 | 32.1 | 81.7 | + | + 0.66 |
| 1DS-1 | 9.1 | 22.6 | 45.2* | 27.5* | 86.9 | + | + 0.59 |
| 1DS-2 | 8.9 | 22.2 | 55.6 | 33.4 | 89.1 | + | + 0.57 |
| 1DS-3 | 8.8 | 21.8 | 55.4 | 31.9 | 86.7 | + | + 0.48 |
| Dt1DL | 7.5* | 20.0 | 31.6* | 27.2* | 57.3* | + | Cen -1.0↓ |
| 1DL-4 | 10.4** | 21.2 | 19.0** | 24.9** | 51.7** | − | − 0.18 |
| 1DL-1 | 11.4** | 22.0 | 22.6** | 19.9** | 44.4** | − | − 0.23 |
| 1DL-3 | 11.2** | 22.4 | 30.8** | 21.8** | 47.0** | − | − 0.25 |
| 1DL-6 | 11.0** | 21.4 | 18.6** | 22.9** | 39.0** | − | − 0.29 |
| 1DL-8* | 8.5 | 22.4 | 72.0* | 31.6 | 91.5 | + | − 0.33 |

1DL-8* designated as CSDt1DSAL, *CS* Chinese Spring, *TGW* thousand-grain weight.
The degree of significance indicated as **P*, 0.05; ***P*, 0.01; *n* = 5 biologically independent samples.
[a]Viable (+) and non-viable (−) hybrid seeds in cross with rye.
[b]Breakpoint intervals according to the data from
https://www.k-state.edu/wgrc/genetic_resources/deletion_lines/group_1.html

(statistically significant at $P < 0.01$) number of grains per spike, seeds set and thousand-grain weight in comparison to line 1DL-8, CS, and the 1DS deletion lines. In addition, deletion lines 1DL-4, 1DL-1, 1DL-3, and 1DL-6 have significantly (at $P < 0.01$) longer spikes (Fig. 1, Table 1). Thus, a new locus has a pleiotropic effect on spike morphology and productivity, or other loci located in the target area are involved in the regulation of spike morphology and fertility in hexaploid wheat.

To confirm the genetic structure, all lines were genotyped by genotyping by sequencing (GBS) (Supplementary Fig. 2, Supplementary Fig. 3, Supplementary Table 1). GBS analysis revealed that line 1DL-8 is characterized by an excess of chromosome 1D short arm sequences. Subsequent FISH analysis demonstrated that 1DL-8 is an additional line containing besides the full set of 1D chromosomes two additional short arms of 1D (Supplementary Fig. 4). In the following, this line will, therefore, be designated as CSDt1DSAL.

To complete the physical mapping of the new locus we tested line 1DL-2 which has a breakpoint interval −0.41 on the long

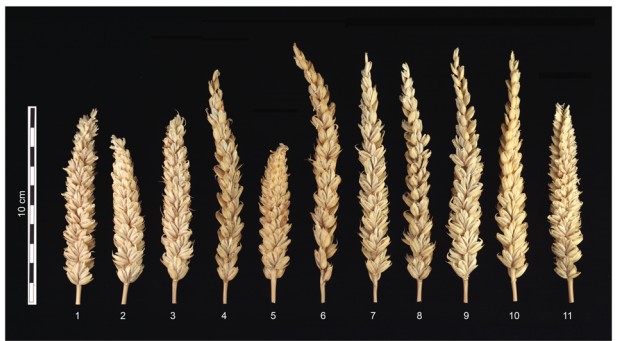

**Fig. 1 Spike morphology in wheat lines.** Lines: (1) CS, (2) N1D/T1A, (3) N1D/T1B, (4) Dt1DS, (5) Dt1DL. (6) 1DL-4. (7) 1DL-1. (8) 1DL-3. (9) 1DL-6. (10) 1DL-2, (11) CSDt1DSAL (1DL-8).

arm of chromosome 1D under field and greenhouse conditions (Supplementary Table 2a, b) for the ability to produce viable seeds in crosses with rye and for analyzing its productivity and spike structure. The analysis showed that line 1DL-2 gives rise to similar changes in spike morphology (Fig. 1), as recorded for the deletion lines 1DL-4, 1DL-1, 1DL-3, and 1DL-6, which have chromosome 1D long arm deletions. Both fertility and thousand grain weight were extremely reduced in these deletion lines, especially under field conditions (Supplementary Table 2a, b). More detailed phenotypic data of the main spike characteristics in hexaploid wheat under different growing conditions are available in Supplementary Data 1. 1DL-2 is not able to form viable hybrid seeds in crosses with rye as well as lines N1D/T1A, Dt1DS, and the other four 1DL lines. Because the 1DL-2 has a −0.41 breakpoint interval on the long arm of chromosome 1D, our locus (loci) of interest should be located below this interval. In the CS wheat genome, the interval of deletion that includes the locus is located within the interval 259,000,000 bp to 494,000,000 bp on chromosome arm 1DL. A large number of genes are located within this interval (5655 genes) of which 2650 genes are high confidence (HC) genes that make it unattainable to suggest a candidate gene. The list of candidate genes within the physical deletion interval in the CS genome is available in Supplementary Data 2. This locus is named *DEFECTIVE ENDOSPERM–D1* (*Dee-D1*). The homoeologous locus located on chromosome 1B and partially compensating for the absence of chromosome 1D is named *Dee-B1*.

**Endosperm and embryo development.** Embryo and endosperm development were studied in both isolated embryo sacs (ESs) and by using histological medial transverse sections through the caryopsis. At 3 days after pollination (DAP), there were significant differences in endosperm development between the crosses CS × L6 and N1D/T1A × L6 (Fig. 2a, c). Furthermore, 3 DAP embryo sacs in caryopses from cross CS × L6 displayed a large central vacuole surrounded by a multinucleate syncytium at

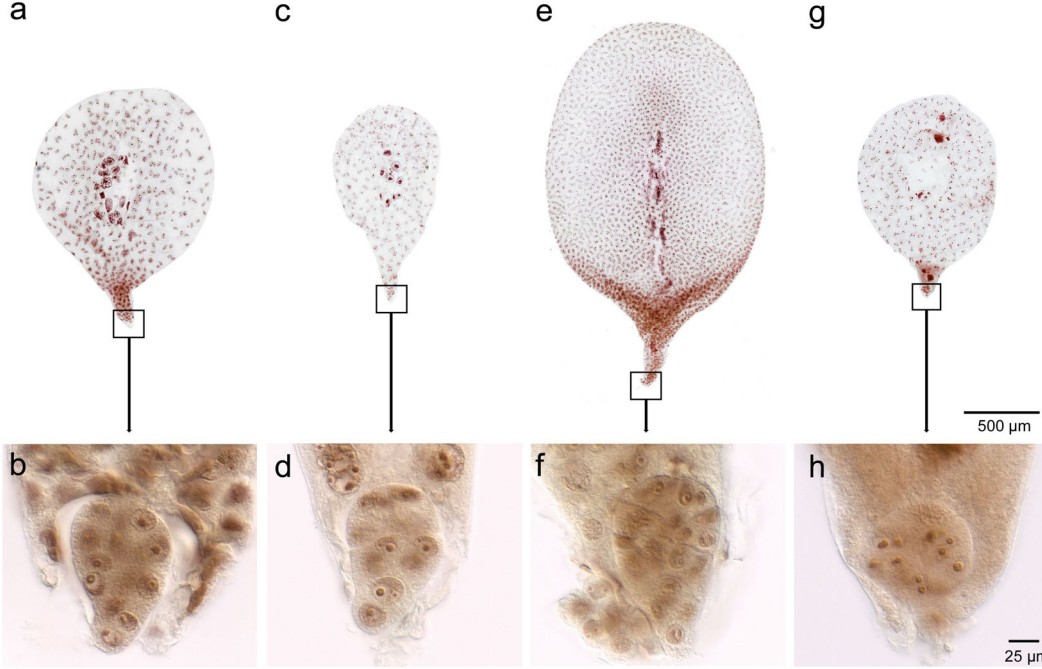

**Fig. 2 Embryo and endosperm development of wheat x rye crosses. a, b** CS x L6, 3 DAP. **c, d** CS N1D/T1A x L6, 3 DAP. **e, f** CS x L6, 5 DAP. **g, h** CS N1D/T1A x L6, 5 DAP, isolated embryo sacs. While between 3 DAP and 5 DAP the embryo sac of CS x L6 significantly increases (**a, e**), that of N1D/T1A x L6 hardly increased size (**c, g**). Detail recordings of the embryo show that at 3 DAP embryos of both crosses appeared similar (**b, d**). At 5 DAP the embryo of CS x L6 grew in size (**f**) while that of N1D/T1A did not grown and showed clear signs of nuclear degradation.

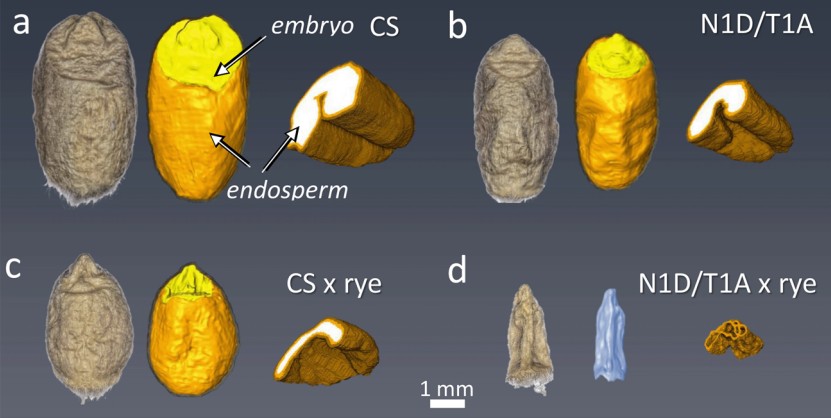

**Fig. 3 Structure of intact mature grains as revealed by comparative 3D-MRI analysis.** (**a**) CS self-pollination, (**b**) CS N1D/T1A self-pollination; (**c**) CS x rye; (**d**) CS N1D/T1A x rye. From left to right: the image of intact seed, 3D-model of embryo and endosperm; virtual cross-section showing the shape of endosperm (white) or empty space (drawn in blue) inside of the grain. See also Movie S1.

the periphery (Fig. 2a). At this stage, the lemon-shaped embryos are comprised of 14–16 cells (Fig. 2b). Endosperm cellularization started from the micropylar end of the ES at 3 DAP and was completed at 5 DAP (Fig. 2e). During this stage, the ES had markedly increased in size and the embryo contained 28–32 cells (2f). The fertilized ES development of N1D/T1A × L6 hybrids initially proceeded similar to that in the control crosses. Although at 3 DAP the ES of N1D/T1A × L6 was smaller in size than that of CS × L6, the hybrid embryos of both contained 12–14 cells (Fig. 2d). The primary endosperm nucleus displayed a normal-looking peripheral syncytium (Fig. 2c). There was no sign of cell-wall formation at the micropylar end of the ES (Fig. 2d). Two days later, at 5 DAP the ES of N1D/T1A × L6 had only slightly increased in size and the endosperm was still syncytial with no sign of cell-wall formation. Furthermore, the hybrid embryo had developed no further and showed signs of nuclear degeneration which was also apparent in the micropylar end of the endosperm syncytium (Fig. 2g, h).

No significant differences in endosperm development of CS × CS, CS × L6, and N1D/T1A × L6 were found in transverse sections through the middle region of the developing caryopsis at 2 DAP (Supplementary Fig. 5a–c). At 5 DAP, caryopsis diameter of the CS × CS and CS × L6 hybrids had increased significantly, and the endosperm had become cellularized (Supplementary Fig. 5d, e). In the N1D/T1A × L6 cross, however, caryopsis diameter did not increase between 2 DAP to 5 DAP and endosperm did not cellularize (Supplementary Fig. 5f).

**The structure of mature grains.** Nuclear magnetic resonance imaging (MRI) analysis was performed for 11 out of 16 studied lines and five hybrid combinations from crosses CS, N1D/T1A, Dt1DS, 1DL-6, CSDt1DSAL with rye (Supplementary Fig. 6). An exception was made for five lines carrying deletions of the short arm of chromosome 1D since they did not reveal critical abnormalities in the development of the endosperm and embryo. The grains of studied lines substantially varied in size and shape due to differences in endosperm/embryo volume. Hexaploid wheat CS and line N1D/T1A produced grains with similar shapes, but the embryo and endosperm volume of the N1D/T1A line was about 2.5-fold smaller as compared to CS (Fig. 3; Supplementary Fig. 6). Crosses of both lines with rye resulted in the loss of grain volume: 66% in CS × rye and 80% in N1D/T1A × rye (Fig. 3c, d). Endosperm in CS × rye was horseshoe-like shaped and up to 5 times smaller as compared to CS (Supplementary Fig. 6). The crossing of N1D/T1A × rye resulted in empty grains with an

obvious lack of both embryo and endosperm (Fig. 3d; Supplementary Movie 1). The absence of both embryo and endosperm in mature kernels explains the viability loss of N1D/T1A × rye hybrid seeds (Table 1). Importantly, neither embryo nor endosperm was present in mature grains of crosses N1D/T1A, Dt1DS, 1DL-6 with rye (Supplementary Fig. 6). The empty space formed inside of caryopsis due to the early degradation of embryo and endosperm (Fig. 2; Supplementary Movie 1) caused shrinkage of pericarp during drying. The pericarp was the only organ contributing to the dry weight of caryopsis at maturity.

**Genotypic variation in floret fertility-related traits.** Analysis of spikelet productivity of the main culm spike revealed that deletion lines in which the *Dee-D1* locus absents (Dt1DS, 1DL-1, 1DL-2, 1DL-3, 1DL-4, 1DL-6) show reduced fertility in the central part of the spike and complete sterility in the upper third of the spike (Supplementary Fig. 7a–d). Under field conditions, deletion lines still produced a few seeds (7.8–23.1%) in the apical part of the spike, whereas under greenhouse conditions, the upper 5–6 spikelets are always underdeveloped and sterile. Wheat has the ability to form more spikelets per spike under unrestricted feeding (31.6 spikelets in CS under greenhouse compared to 24.2 spikelets under field condition, Supplementary Table 2a, b) which for the most part always remained sterile. Under both greenhouse and field conditions, the productivity of line Dt1DL was significantly less than that of CS. Still, line Dt1DL showed the same spike morphology (Fig. 1) and produced viable hybrid seeds in crosses with rye (Table 1, Supplementary Table 2a, b, d). This suggests that the short arm of chromosome 1D also carries loci with a significant influence on spike productivity in hexaploid wheat.

Crucial changes in spike morphology and fertility in lines with the absence of *Dee-D1* prompted us to conduct a detailed analysis of the formation of floral meristem and the implementation of grain productivity in two lines containing *Dee-D1* (CS, CSDt1DSAL) against three lines in which *Dee-D1* is absent (Dt1DS, 1DL-6, and N1D/T1A). At the terminal spikelet stage, spikes of tested lines were 18–20 mm long and there were no visible differences in spike morphology between lines with or without *Dee-D1* (Fig. 4). N1D/T1A line formed significantly ($p <$ 0.01) more spikelets per spike (28.8 in 2016 and 31.4 in 2017, Supplementary Table 2c, d) compared to CS (26.0 and 25.2, respectively). This higher spikelet number, however, did not increase grain productivity. The number of floral primordia per spikelet, fertile floret number per spikelet, and the number of grains per spikelet showed significant variation between spikelet

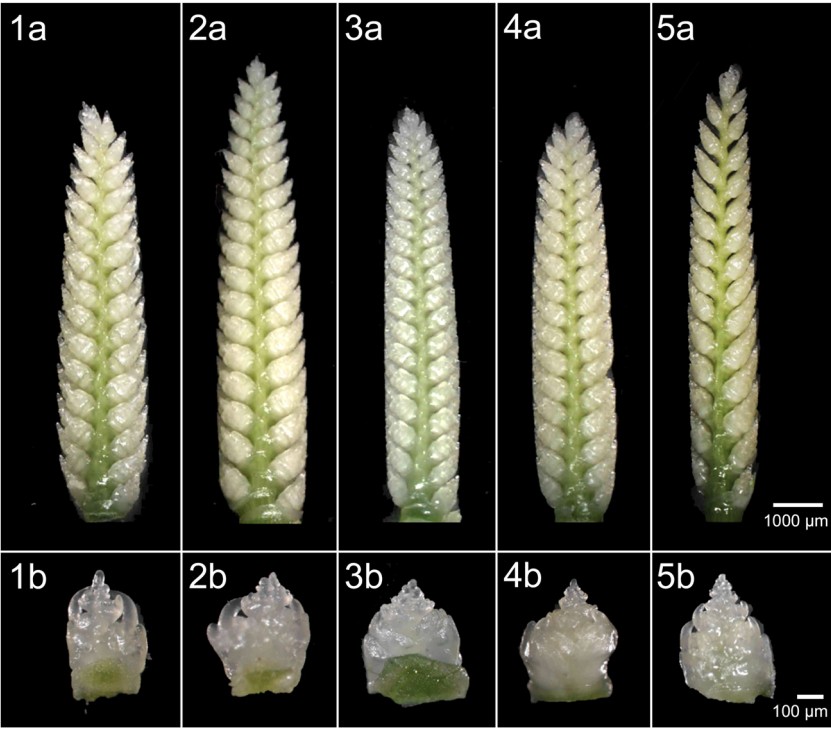

**Fig. 4 Morphology of spikes and spikelet at terminal spikelet (TS) stage.** The spike morphology at TS in line (**1a**) CS, (**2a**) N1D/T1A, (**3a**) Dt1DS, (**4a**) 1DL-6, (**5a**) CSDt1DSAL and at the basal spikelets in line (**1b**) CS, (**2b**) N1D/T1A, (**3b**) Dt1DS, (**4b**) 1DL-6, (**5b**) CSDt1DSAL.

positions along the spikes studied. The number of fertile florets per spikelet was strongly associated with floret survival and only weakly related to the number of floral primordia per spikelet (Supplementary Fig. 8; Supplementary Table 3). The Dt1DS line formed significantly ($P < 0.01$) more floral primordia per spikelet in basal and central parts of the main culm spike than other lines (Supplementary Fig. 8c–f), but at the same time the number of flowers and the ability to form grains (measured as number of grains per spikelet) was one of the lowest (Supplementary Fig. 8c–f). Changes in spike morphology and the abortion/survival of fertile florets took place between the terminal spikelet and boot swollen stages. Strong changes occurred in the lines Dt1DS and 1DL-6, but not in line N1D/T1A in which the *Dee-D1* absent too. Potential productivity in the main culm spike was found to depend on the positions of the spikelet and the presence of the *Dee-D1*. In CS and CSDt1DSAL line, 33.3–45.5% of floral primordia in basal spikelets and 30.0–40.0% of those in central spikelets eventually produced grains (Supplementary Table 3). In the lines lacking *Dee-D1* potential productivity ranged from 20.1–30.6% in basal spikelets to 8.1–25.3% in the central spikelets (Supplementary Table 3). Regardless of genotype floral primordia of the apical spikelets rarely give rise to seeds (0.0–11.4%, Supplementary Table 3). In all tested lines with *Dee-D1*, excluding the Dt1DL line, at harvest maturity stage both under greenhouse and field conditions (2016–2018) 72.3–86.4% florets per spikelet in basal and 68.6–79.2% in the central part of the main culm spike form grains. In all lines, apical spikelets were mostly sterile (Supplementary Fig. 7a–d).

## Discussion

Using a series of nullisomic-tetrasomic and chromosome 1D deletion lines of hexaploid wheat CS, we identified the exceptional role of chromosome 1D in endosperm formation both for bread wheat and in crosses with rye. Our experiments with chromosome 1D deletion lines proved that the short and long arms of

this chromosome have their own special contributions to spike productivity. The long arm of chromosome 1D carries the newly identified *Dee-D1* locus, whose loss abolishes the formation of the endosperm in both bread wheat and intergeneric hybrids of wheat. Examination of the Dt1DS line and 1DL-1, 1DL-4, 1DL-3, 1DL-6, 1DL-2 lines showed that the absence of *Dee-D1* reduces the number of grains per spike by 25–30% and thousand grain weight by at least 10% (Table 1; Supplementary Table 2a–d) both under field and greenhouse conditions. The crosses between N1D/T1A, Dt1DS and deletion lines for 1DL with rye are leading to the formation of "empty" grains without traces of endosperm and embryo (Fig. 3; Supplementary Fig. 6; Supplementary Movie 1). Tetrasomy for chromosome 1B partially compensated for the absence of 1D for spike productivity in wheat (Table 1; Supplementary Table 2a) and the intergeneric crosses with rye[13]. This confirms the presence of a homoeologous locus on chromosome 1B, which we named *Dee-B1*. Compared to the CS control, the lines Dt1DS, 1DL-1, 1DL-4, 1DL-3, 1DL-6, and 1DL-2 have an altered spike morphology (Fig. 1) and impaired spike fertility (seeds set). This may represent the pleiotropic effects of the *Dee-D1* or is the result of the absence of other important loci located in the same targeted region and involved in the regulation of spike morphology and spike fertility in hexaploid wheat.

The *Dee-D1* seems to exert its effect in the later stages of spike development. Apart from a larger number of floral primordia per spikelet measured in the Dt1DS line in the terminal spikelet stage (Fig. 4), floral meristem formation was not affected in the deletion lines until after the terminal spikelet stage, eventually leading to a sharp decrease in the number of grains and thousand-grain weight (Table 1, Supplementary Table 2a–d). The result demonstrated that 1DS carries locus that significantly influenced spike productivity in hexaploid wheat, and the suggested locus has negative impact on floret fertility-related traits. Our results indicate that the long arm of chromosome 1D low breakpoint interval – 0.41 (Supplementary Table 2a) carries a locus that plays a crucial role not only in the endosperm development in wheat-

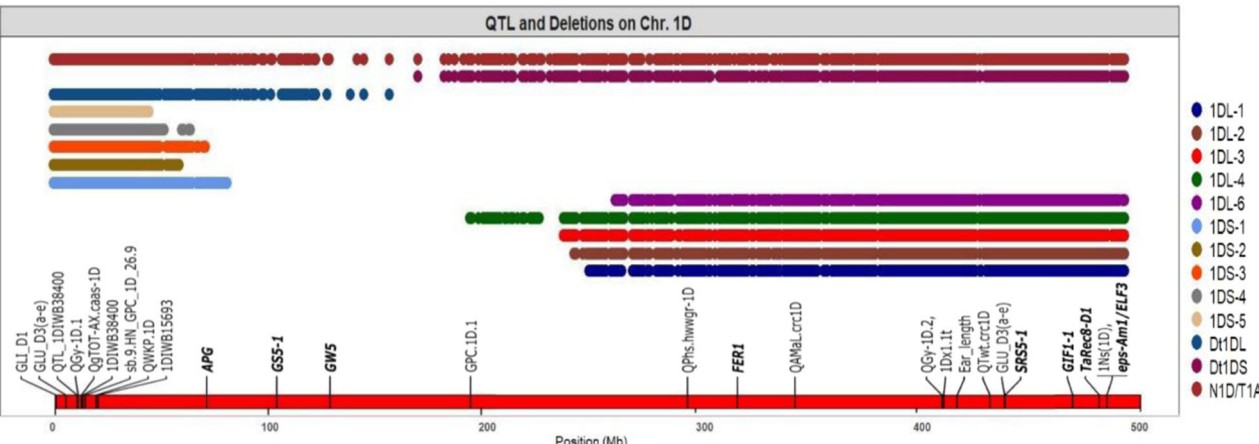

**Fig. 5 Physical mapping of chromosomal and/or segmental deletion in lines at 1D chromosome.** Dots denote the deletion position in the mutants according to GBS reads within 1 Mb in a euploid CS. QTL and genes were physically mapped using IWGSC Ref Seq v1.1 wheat genome assembly[8].

rye hybrid grains but also in endosperm formation of hexaploid wheat, which is itself a natural amphidiploid. The absence of the *Dee-D1* leads to (i) the formation of shriveled grains and (ii) the formation of non-viable grains in crosses with rye. This suggests that *Dee-D1* can be classified as a species-specific locus with positive effects on the fate of newly formed polyploid organisms.

Important quantitative trait loci (QTLs) have been found on all chromosomes of bread wheat[14]. Li and Yang[15] identified six potential GS candidate genes on chromosome 1D of the wheat genome involved in yield formation. According to the literature, the short arm of the 1D chromosome includes clusters of QTLs e.g. *QGy-1D.1, QWKP.1D, 1DIWB38400* and *1DIWB15693* which are located on the terminal part of 1DS and the three genes *ANTAGONIST OF POSITIVE REGULATOR OF GRAIN LENGTH (APG), GRAIN SIZE5-1 (GS5-1)*, and *GRAIN WEIGHT 5 (GW5)* which are located more towards the centromeric region (Fig. 5). QTLs from a terminal cluster are underlying the grain yield-related traits improvement through thousand-grain weight[16] and grain weight[17,18]. Five lines (1DS-1 to 1DS-5) carrying the deletion at the short arm (Fig. 5) did not show differences in the studied traits compared with CS. The finding suggested that the absence of this cluster of QTLs and *APG* gene in the deletion lines at 1DS did not influence critically the productivity of the main culm spike in CS when *Dee-D1* is present on 1DL. Two other important genes involved in grain size and weight in wheat are physically mapped on 1DS: *GS5-1* and *GW5* according to Li and Yang[15]. In rice, *GS5* that encodes a putative serine carboxypeptidase acts as a positive regulator of grain size by regulating grain width, filling, and weight as a consequence of increasing the grain size that could be attributed to an increase in cell number[19] whereas *GW5* is a positive regulator of brassinosteroid signaling pathway and has positive effects on grain width and weight[20]. The deletion in *GW5* in rice influences grain width and weight through reducing expression levels of *GW5* in the panicle. The Dt1DL line is the carrier of the *Dee-D1*, which ensures the development of viable hybrid seeds in crosses with rye, but would probably not compensate for the absence of *GS5-1* and *GW5* genes, therefore, it has significantly lower productivity in all measured traits (Table 1; Supplementary Table 2a, b, d). The CSDt1DSAL line carries two additional copies of 1DS and, respectively, a double dose of *GS5-1* and *GW5* genes. In this line, however, the number of grains was higher than in CS and thousand-grain weight similar in both under greenhouse (Supplementary Table 2c, d) and field conditions (Table 1). This indicates that the presence of two additional copies of *GS5-1* and *GW5* leads to a higher number of grains per spike than in CS itself.

The long arm of chromosome 1DL carries QTLs, known genes (Fig. 5) and *Dee- D1* from pericentromeric to telomeric regions which have a strong impact on spike morphology, grain number, thousand-grain weight and seeds set for hexaploid wheat. This region includes grain protein content (GPC) QTL (*qGPC.1D.1*) that showed a negative impact on protein content in wheat grains[21]. The locus qGPC.1D.1 is located upstream of the −0.41 breakpoint interval and present in 1DL-1, 1DL-2, 1DL-3, and 1DL-6. In line 1DL-4 this region is deleted. All these lines do not have significant differences in the studied grain traits between each other and, therefore, the presence or absence of qGPC.1D.1 cannot be associated with the observed changes (Table 1; Supplementary Table 2a, b, d). Besides qGPC.1D.1 the deletion region downstream of the −0.41 breakpoint interval harbors much important grain yield and spike morphology related QTLs and genes[15]. *FERONIA (FER)* is an Arabidopsis gene inhibiting integument cell elongation or endosperm division and thereby seed size[22]. The *FER* ortholog gene in rice is *DWARF AND RUNTISH SPIKELET1* and 2. Further genes (*DRUS1* and *DRUS2*) control reproductive growth and development by maintaining cell viability and repressing cell degradation[23]. *Small and round seed 5 (SRS5)* is another important gene that reduced cell number, inhibits cell elongation, and division leading to smaller seeds and shorter panicle in rice[24]. This gene is deleted in all tested 1DL deletion lines and, therefore, may cause a reduction in seed number, thousand-grain weight, and seeds set in addition to altering spike shape. According to Li and Yang[15], *GRAIN INCOMPLETE FILLING 1 (GIF1)* is also located on 1DL that encodes a cell-wall invertase and is known to be involved in shortening grain filling duration and then a reduction in grain weight[25]. Substitution of 1Ns chromosome in Leymus mollis Trin. (a wild relative of common wheat) by chromosome 1D (1Ns (1D)) improve wheat yield by increasing florets and spikelets number, grain weight, spike length and thousand-grain weight[26]. *EARLY FLOWERING 3 (ELF3)*, a candidate for the earliness per se locus *Eps-Am* 1 in wheat[27,28], plays a crucial role in flowering time, which in turn influences development and grain yield. Suggesting that the deleted region contains this gene an early flowering leads to fewer grains and thousand-grain weight.

However, in the scientific literature, there is no information about genes/QTLs for chromosome 1D leading to the formation of postzygotic barriers in distant crosses of hexaploid wheat with rye as a result of impaired development of hybrid endosperm. All genes and QTLs mentioned above should not be related to *Dee-D1*. According to the physical interval of deletion at 1DL in the CS wheat genome, the interval harbored 5655 genes whereas 2650

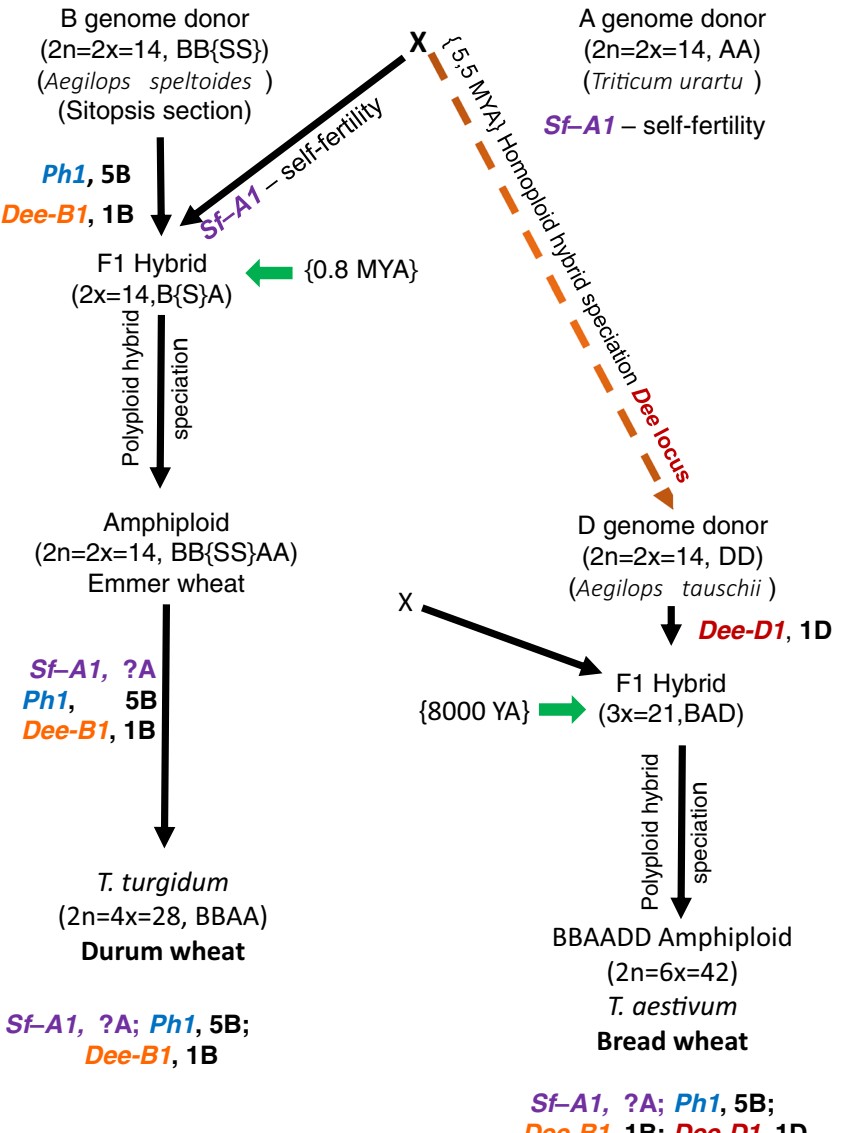

**Fig. 6 Putative involvement of *Dee-D1* in hybridization events.** *Dee-D1* leads to the evolution of tetraploid and hexaploid wheat, the evolution model based on Mujeeb-Kazi et al.[48] and Rasheed et al.[30].

HC genes therefore we did not succeed in designating candidate genes for endosperm development and floret/spikelet fertility in wheat. However, this study identified a promising novel locus for endosperm development and floret/spikelet fertility in hexaploid wheat that needs further functional analysis.

In our concept of the evolution of wheat, we propose that the emergence of the tetraploid form was the result of combining in one genotype of the progenitor of B genome at least two major speciation loci with positive effect: (i) *Ph1*, localized on chromosome 5B, providing the bivalent pairing between the homologous chromosomes during meiosis[29] and (ii) *Dee-B1*, localized on chromosome 1B, regulating the normal development of the endosperm in the newly formed hybrid organism[13]. The role of the A genome in the evolution of the first tetraploid and eventually hexaploid wheat was the introduction of a dominant locus, which we have tentatively named *Sf-A1*. This locus provided self-fertility to the first tetraploid wheat.

From an evolutionary point of view, the hybridization of *T. turgidum* with a rare form of *Ae. tauschii*[30] carrying *Dee-D1*, was another revolutionary event in the evolution of the genus *Triticum*. Since the presence of this locus is a need and prerequisite for

successful distant hybridization, we can assume that only those samples of *Ae. tauschii*, which show high crossbreeding with tetraploid wheat (less than 1% of 372 tested crosses[31]), carry this locus. In the remaining samples of *Ae. tauschii* this locus may be either absent or altered, for example, close to the structure of the *Dee-B1* (Fig. 6; Supplementary Fig. 9), which leads to the death of the hybrid endosperm in distant crosses in the early stages of the development of the hybrid caryopses.

The probability that hexaploid wheat could have emerged several times is quite small, since it requires multiple rare events to coincide, including simultaneous flowering of *T. turgidum* and *Ae. tauschii*; the formation of unreduced gametes in the parental species, or a high probability of nondisjunction of the parental chromosomes in the first division of the zygote; and the mandatory presence of locus *Dee-D1* in the genotype of *Ae. tauschii*, in order to regulate the competitive relationship between the homoeologous genes of the parental genomes responsible for the development of the endosperm in the hybrid caryopsis.

The question of why this locus remained elusive for so long may lay in its ubiquity. Due to its important function, this locus should be extremely conserved and identical in all selected and

commercial varieties. Any mutations leading to a decrease in spike productivity is automatically subjected to negative selection either by breeders or by nature (natural selection). Evidence in favor of this assumption was brought forward by Chen et al.[32]. Using genome-wide association analysis, the genetic diversity, population structure, and LD decay were estimated in a winter wheat association mapping panel ($n = 205$) and markers associated with thousand-grain weight and related traits were identified. This analysis showed that the selected wheat varieties showed no genetic differences in terms of spike productivity, and no QTLs for these traits were mapped on chromosome 1D. This suggests that the structure of the responsible locus was identical in all varieties analyzed.

The rapid success of bread wheat as a new species and as a product of distant natural hybridization was the result of at least four species-specific loci with positive effect within the new hybrid genome: Sf-A1, Ph1, Dee-B1, and Dee-D1 (Fig. 6). These loci were capable to coordinate and resolve conflicts between the genes of the parental genomes within different genetic systems. From a population point of view, this new allopolyploid species showed increased fitness compared to its parents. This enabled T. aestivum to colonize new niches[5] and to become the platform for the emergence of the new genus Triticale (X Triticosecale Wittmack)[33], and further evolutionary transformations of hexaploid wheat at intra-specific and intergeneric levels[33,34].

Further study of the Dee-D1 phenomenon may help to understand the precise integration of various events within the genetic program leading to the endosperm formation in allopolyploid plants. The features that characterize this unique developmental pathway comprise fundamental information essential to both the evolution of hexaploid wheat and the identification of new genetic regulators. It might help to improve endosperm development (grain yield) in polyploid wheat, which is a world important biological resource for sustaining the human population.

Overall, we discovered and mapped a novel DEFECTIVE ENDOSPERM–D1 (Dee-D1) locus that is physically located on the long arm of chromosome 1D and has never been studied before. Dee-D1 plays a crucial role in endosperm formation in hexaploid wheat. The absence of Dee-D1 leads to produce shriveled grains and the formation of non-viable grains in crosses with rye. The absence of Dee-D1 in the genome of hexaploid wheat leads to a decrease in the number of grains and thousand-grain weight. The loss of fertility and the change in the morphology of the spike can be caused either by other loci located in the target area or the pleiotropic effect of Dee-D1 are involved in the regulation of spike morphology and spike fertility in hexaploid wheat. Dee-D1 and Dee-B1 can be classified as speciation loci with a positive effect on the function of genes that are involved in endosperm development in the hybrid genome. The presence of such loci is necessary for the normal development of endosperm in tetraploid and hexaploid wheat. Dee-D1 has a greater force of action than Dee-B1 in the genome of hexaploid wheat and normalizes the development of hybrid endosperm in distant hybrids. Finally, the short arm of chromosome 1D carries the well-known genes GS5-1 and GW5 which are influenced on seed set and seed size in the spike of hexaploid wheat. The presence of these genes in the Dt1DS line and 5 deletion lines along the long arm does not compensate for the absence of Dee-D1.

## Methods

**Plant material and growth conditions.** Hexaploid wheat cv. Chinese Spring (CS) and two nullisomic-tetrasomic lines, N1D/T1A, N1D/T1B were supplied by the John Innes Centre, Norwich, UK, whereas two ditelosomic lines, Dt1DS, Dt1DL and 10 deletion lines (1DS-1, 1DS-2, 1DS-3, 1DS-4, 1DS-5, 1DL-1, 1DL-3, 1DL-4, 1DL-6 and 1DL-8) for chromosome 1D of CS were kindly provided by Dr. W. Jon

Raupp (Wheat Genetic & Genomic Resources Centre, Kansas State University, Manhattan, USA). Line CSDt1DSAL was selected in this study from 1DL-8 line via GBS and FISH analysis. Plants of these lines were grown under both field and greenhouse conditions. Lines were evaluated under field conditions in 2015. Plants of CS, N1D/T1A, Dt1DS, 1DL-6, CSDt1DSAL lines were grown in the greenhouse during 2016 while lines CS, N1D/T1A, Dt1DS, Dt1DL, 1DL-4, 1DL-1, 1DL-3, 1DL-6, CSDt1DSAL were grown in the greenhouse during 2017. To complete the physical mapping of the new locus, the Wheat Genetic & Genomic Resources Centre of Kansas State University provided line 1DL-2 which has a breakpoint interval −0.41 on the long arm of chromosome 1D, afterward. This line (1DL-2) was tested in 2018 both under field and greenhouse conditions in addition to all other lines for the ability to produce viable kernels in crosses with rye and for analyzing its productivity and spike structure.

Experiments were conducted at the Leibniz Institute of Plant Genetics and Crop Plant Research (Gatersleben, Germany; 51°49′23″ N, 11017′13″ E, altitude 112 m, black soil of clayey loam type, 9 °C average annual temperature and 490 mm average annual rainfall). Plants were grown under field conditions in 2015 and 2018 and greenhouse conditions in 2016, 2017 and 2018. For field experiments, 10 seeds of each line were sown in 96-well trays and kept under greenhouse conditions (16/8 h day/night; ~20/~16 °C) for 21days. On 13 April 2015 and 24 April 2018 plants at the tillering stage (Z21-22)[35] were transplanted into the field, 10 plants per row of 100 cm with 30 cm between rows. Plants were manually irrigated and standard agronomic practices were applied. At the harvest maturity stage, five traits of main culm spike (Table 1) were measured: spike length (cm), number of spikelets, number of grains, weight (g) of 1000 grains (thousand-grain weight) and seed set (SS in %) in florets 1 (F1) and 2 (F2) of each developed spikelet. Reduced spikelets were not included in the fertility analysis.

For greenhouse experiments, germinated seeds of each line were planted into 14 cm pots (one seed per pot) and grown under 16/8 h day/night; ~20/~16 °C until harvest maturity stage. In the 2016 and 2017 seasons, 20 plants for each line were used to study floret primordial initiation, next to floret and spike development. In 2018 a comparative study of spike fertility and productivity included all 16 lines involving 10 plants per line grown under similar greenhouse conditions.

To obtain wheat-rye hybrids, wheat spikes were emasculated 1–2 days before anthesis and pollinated 2–4 days later with fresh rye or CS (control) pollen. For each line at least 100 florets were used. Commercial rye varieties ("Kalinka", "Plato", "Warko", "Motto") from the IPK collection and rye inbred line L6, kindly provided by Voylokov[36] from the rye collection of the St. Petersburg State University, Russia, were used as pollen donors.

**Phenotypic staging and measurements.** In 2016 and 2017 floral development and seed productivity of plants grown under greenhouse conditions were investigated at the following stages: (1) terminal spikelet stage when spikelet initiation was completed[37]; (2) boot swollen stage (Z45) when the developing head was pushed through the full-grown flag leaf sheath; (3) anthesis stage (Z65) when 50% of spikes were in anthesis; and (4) harvest maturity stage (Z92) when plants were ready for harvest[35].

Analysis of floret primordia initiation in spikelets was according to Guo et al.[38] with modifications. For each line, three plants were randomly selected to measure floral primordia per spikelet and potential fertile floret number per spikelet, while main shoots of additional five plants were used for determining the number of flowers and number of grains per spikelet in the harvest maturity stage. All these parameters were determined in six spikelets collected at the basal (third spikelet from the basal part of spike), central (spikelet in the center of spike), and apical (third spikelet from the top) parts on either side of the spike. To determine the number of fertile florets per spikelet at the boot swollen and anthesis stages, aborted florets were distinguished from fertile florets because of their smaller size and the presence of dry or transparent anthers. Potential spikelet productivity in different positions of the spike was determined as the ratio of the average number of grains in spikelets at the harvest maturity stage compared to the average number of floret primordia in spikelets in the same position at the terminal spikelet stage. Collected data were subjected to analysis of variance (ANOVA) at the probability level $P \leq 0.05$, $P \leq 0.01$, and $P \leq 0.001$ using GENSTAT for Windows ver. 19 (VSN International, Hemel Hempstead, UK). Student's t test with one-way ANOVA combined with Fisher's Least Significant Difference (LSD) test was used to compare among lines of each trait at the same developmental stage from the same experiment.

**Cytological and histological analysis of the endosperm and embryo development.** For a detailed analysis of endosperm and embryo development whole embryo sacs (ES) of the crosses CS × CS, CS × L6, N1D/T1A × N1D/T1A and N1D/T1A × L6 were isolated after 2, 3, and 5 days after pollination DAP. For each cross and time point 10 caryopses were fixed in Chamberlain's fixative, ESs were extracted and stained as previously described[39].

For histological analysis of whole caryopses, 10 caryopses at 2, 3, and 5 DAP were fixed overnight at 8 °C with 1% glutaraldehyde and 2% formaldehyde in phosphate buffer (50 mM, pH 7.0). To facilitate infiltration, a small slice was cut from the apical part of the caryopses. After fixation probes were briefly washed with buffer (1×) and distilled water (2×), dehydrated in a graded ethanol series, and embedded in Spurr's low-viscosity resin. Medial transverse semithin 1 µm sections

were cut on a Reichert-Jung Ultracut S microtome (Leica, Vienna, Austria) and stained with crystal violet. Probes were examined in a Zeiss Axiovert microscope (Carl Zeiss, Jena, Germany).

**Fluorescence in situ hybridization**. Fluorescence in situ hybridization (FISH) was carried out for the 1DL-8 deletion line to confirm the karyotype structure of this line. Seeds were germinated on moistened filter paper in a glass Petri dish in the dark at 22–25 °C for 2–3 days. Root tips were pre-treated in 1.5 ml Eppendorf tube with ice-water for 24 h at 4 °C. Subsequently, the root tips were fixed in 90% acetic acid for 5 min followed by washes in distilled water. Preparation of mitotic chromosomes was performed according to Han et al.[40] by dropping cell suspensions on the slides which were then stored until use in 96% ethanol at 4 °C. FISH with oligo-119.2 and oligo-535 probes were carried out to distinguish wheat homoeologous chromosome[41]. FISH experiments were performed following the protocol of Wu et al.[42] with minor modifications. In brief, after 5 min washing in 2 × saline sodium citrate buffer (SSC), slides were treated with 45% acetic acid for 10 min, washed in 2 × SSC for 5 min twice and fixed in freshly prepared 4% formaldehyde for 10 min. After washing in 2 × SSC three times for 5 min, slides were dehydrated in ethanol series (70, 85, 96%) for 2 min in each and dried at room temperature. After adding 10 μl of hybridization cocktail slides were covered with a coverslip. After 90 s incubation on a hot plate set at 80 °C slides were incubated in a moist chamber at 37 °C overnight. After removing the coverslips, slides were washed in 2 × SSC at 58 °C for 20 min, then dehydrated with ethanol series, air-dried in the dark and counterstained with DAPI + Vectashield (10 μl per slide). Images were made with Olympus BX61 fluorescence microscope. Images were collected in greyscale and pseudo-colored with Adobe PHOTOSHOP CS (Adobe). Chromosome characterization referred to as Tang et al.[41].

**Analysis of the structure of mature seeds by nuclear MRI**. The 400 MHz Avance nuclear magnetic resonance (NMR) spectrometer (Bruker Biospin, Rheinstetten, Germany) was used for imaging individual grains. High resolution, frequency- selective imaging was performed in vivo. NMR resonators with an inner diameter of 5 mm were employed as the radio frequency (RF) coil. Matrix size and field of view were correspondingly adjusted to achieve a spatial resolution between 50 and 104 μm. In the spin-echo sequence, the repetition times (TR) selected were between 500 and 1000 ms, the echo times (TE) were set to the minimal value, namely between 4.4 and 7.9 ms. To optimize the signal-to-noise ratio, the datasets were averaged two to eight times. Depending on the field of view, an imaging experimental time of 3.5–16 h, was required. Image processing and analysis were performed using MATLAB software (The MathWorks, Natick, MA, USA).

**GBS analysis of wheat lines**. GBS libraries were prepared according to Wendler et al.[43] and sequenced on an Illumina HiSeq2500 instrument (1 × 107 bp) reads at IPK Gatersleben. After adapter trimming with cutadapt, reads were mapped to the Chinese Spring Reference genome sequence assembly (version 1.0[8]) using minimap2[44]. Alignment records were converted to BAM format with SAMtools[45] and sorted according to alignment position with Novosort (http://www.novocraft.com/products/novosort/). The counts of uniquely mapped reads in non-overlapping 1 Mb bins of the Chinese Spring reference were determined with an AWK script and plotted along the genome using plot functions of the R statistical environment (http://www.R-project.org/). Read counts were normalized in two steps. First, raw counts in each genotype were divided by the sum of total read counts in that genotype to account for reading depth differences. Second, log2-ratios between reading counts in each bin between each sample and a Chinese Spring GBS sample were taken to account for coverage differences along the genome due to an uneven distribution of GBS targets. Accession numbers and mapping statistics of GBS samples are reported in Table S1.

**Reporting summary**. Further information on research design is available in the Nature Research Reporting Summary linked to this article.

## Data availability

GBS raw data were deposited in the European Nucleotide Archive (accession: PRJEB37818). Read depth statistics under accessible under https://doi.org/10.5447/ipk/2020/25 in the PGP repository[46]. Locus information on Dee-D1 was deposited at the GrainGenes database[47] under : https://wheat.pw.usda.gov/cgi-bin/GG3/report.cgi?class=locus;name=DEFECTIVE+ENDOSPERM-D1+(Dee-D1)

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

## Acknowledgements

The authors gratefully acknowledge the excellent technical support by S. König, M. Knauft, I. Walde, E. Munz, S. Wagner, A. Marlow, and A. Fiebig for handling raw data submission into the European Nucleotide Archive. We thank Victoria Carollo Blake and Taner Zen for registering Dee-D1 in the GrainGenes database. This study was funded by the German Research Foundation (No. BO 1423/17-1/603175).

## Author contributions

N.T. and A.B. designed the research. N.T. guided the entire study, performed the research and data analysis with help from S.S., M.N., and A.M.A. A.M.A. carried out the statistical analysis, Q.T.L. and gene mapping. L.B. and S.O. carried out the N.M.R. analysis and M.R.I., M.M. and A.x.H. carried out the G.B.S. analysis. T.R. and N.T. carried out the cytological and histological analysis. D.W. and A.n.H. performed the FISH analysis. M.S.R. and M.W.G. discussed the results and revised the manuscript. A.B. initiated the project, guided the study, and revised the manuscript. N.T. and A.M.A. wrote and revised the manuscript with contributions from all coauthors. All authors read and approved the manuscript.

## Funding

## Competing interests

The authors declare no competing interests.
