## [Peer Review File · Communications Biology]

Reviewers' Comments:

Reviewer #1:

Remarks to the Author:

Summary

It is a well-conceived study with the novel finding that the Dee-D1 locus plays an essential role in the endosperm development of hexaploidy wheat and its crosses with rye. Identification of this locus will help the evolutionary geneticist in understanding the evolution of hexaploidy wheat.

The title of the study is clear and concise, and the abstract clearly states the study's objective, methodology, and key findings. The introduction was well-referenced and was not an exhaustive summary of the field. It justifies the choice of wheat to carry out the current study. The design and methodology used in the present study were also appropriate. All information regarding the growth conditions, number of replications, type of data collected, and methods used in the study were mentioned. The deletion lines of chromosome 1D, along with CS and nulli-tetrasomic lines, were tested under both field and greenhouse conditions, and their data was compared accordingly. The results fit with the data provided. In the Discussion section, previous relevant studies are also discussed and cited. For example, findings of the current study are discussed in the light of previously identified grain yield-related QTL's on both arms of chromosome 1D. Moreover, the role of Dee-D1 in spike productivity was also observed. Thus, the concluding remarks are supported by the data.

Minor Comments

- a. The color-coding of different lines in Fig. 6 is not visually separable.
- b. The lines studied for each experiment are clearly mentioned in the result section and the legends of figures and tables. However, a few lines in the Method section stating which lines were studied for what part of the study under what growing conditions will make the flow of reading much better.
- c. Pg 4 line 148, Tables S1a, S1b are probably Tables S2a, S2b
- d. Pg 6, line 216, the role of 1DS in spike productivity was suggested, and later the spike fertility of Dt1DS line was mentioned on Pg 7 (Line 228-231). However, in the discussion section, the role of 1DS was discussed only with respect to the absence of Dee-D1 locus, and no conclusive remarks were made about the presence of 1DS.
- e. Pg 17, Line 579, Mujeeb-Kazi, 2013 reference is missing.

Reviewer #2:

Remarks to the Author:

DEFECTIVE ENDOSPERM-D1 (Dee-D1) locus on the long arm of chromosome 1D seems to be a novel loci for endosperm development in wheat as it was presented by the authors with good evidence and supporting data.

Though the Dee-D1 loci is novel for endosperm development and its associated pleiotropic effect on spike morphology, it seems to be that this loci may be specific for Rye related crosses with hexaploid wheat. Is there any additional crosses made with the 1D deletion lines with hexaploid wheat lines and assessed the effect of this loci?

Secondly the homeolog on chromosome 1B also showed some effect on endosperm development, eventually this should have been tested with few durum based crosses with 1D deletion lines.

Nevertheless, this study identified promising loci for endosperm development in wheat. Apparently, there are more genes regulating the endosperm development in wheat as the volume of endosperm translates into grain yield in wheat and grain yield is governed by several QTL regions.

This manuscript deserves publication in nature communication biology with minor revisions. There are minor corrections needed below:

Line 76: it should be ` world wheat' production

Line 512 to 519, looks like there was a formula but it became like a box, correct this.

Reviewer #3:

Remarks to the Author:

The authors have presented their case for the Dee-D1 loci in wheat present on the short arm of hexaploid wheat's D1 chromosome. This is a novel discovery and the experiment were planned accordingly to make their case. It may be too early to claim this locus and its counterpart on B1 genome as one of the domestication genes. Authors have selected the lines carefully for making the crosses and creating either the addition and/or deletion lines. The phenotypes are contrasting, but they do show some phototropic effects compared to the reference CS line.

Considering that its a genetic loci the data looks sufficient. However as the wheat genome now has a much better quality reference genome, I would like to see which of the genes (sequenced loci) are missing in the region and do any of them or their orthologs from other species have seed/endosperm development and fertility phenotypes. Also check the EMBL-EBI gene expression atlas to see which of the sequenced loci/genes on D1S deleted part show preferred expression in the seed development process to narrow down their candidate genes. Provide all their data.

Submit the GBS data to EMBL-EBI EVA. Authors have submitted the raw data to SRA, but that is not sufficient, they need to submit the SNP calls and GBS data to EVA along with the phenotype data. Need to describe a bit more about the parameters and depth of the sequencing etc.

Reach out to GrainGenes wheat database to register their gene and share the sequencing, GBS and phenotype data for public use. Provide the parental and described lines to the local, national and international seed bank for others to use.

Manuscript title — “*DEFECTIVE ENDOSPERM-D1 (Dee-D1)* is crucial for endosperm development in hexaploid wheat”

Dear Reviewers,

We are grateful to the reviewers for their time and constructive comments on our manuscript. We have implemented your comments and suggestions and resubmitted a revised version of the manuscript for further consideration in the journal. Changes in the initial version of the manuscript are highlighted by YELLOW color in the revised version. Below, we also provide a point-by-point response letter explaining how we have addressed each of the editor or reviewers' comments.

We hope these changes will address the reviewers' comments.

We look forward to receiving a positive response from you and additional comments/corrections, if any.

Yours sincerely,
On behalf of the co-authors
Ahmad Alqudah, PhD

Reviewer's Comments:**In response to the comments by Reviewer 1**

Thank you very much for your review. We have corrected all the suggested comments in the revised version.

Comment:

The color-coding of different lines in Fig. 6 is not visually separable.

Authors' Response:

We considered this comments in the revised Fig. 6 that present new model of wheat evolution with putative involvement of *Dee-D1* in the process.

Comment:

The lines studied for each experiment are clearly mentioned in the result section and the legends of figures and tables. However, a few lines in the Method section stating which lines were studied for what part of the study under what growing conditions will make the flow of reading much better.

Authors' Response:

We have added more description about the used lines in each experiment. Please have a look at lines 431-442.

Comment:

Pg 4 line 148, Tables S1a, S1b are probably Tables S2a, S2b

Authors' Response:

Thank you, we have change it in the text.

Comment:

Pg 6, line 216, the role of 1DS in spike productivity was suggested, and later the spike fertility of Dt1DS line was mentioned on Pg 7 (Line 228-231). However, in the discussion section, the role of 1DS was discussed only with respect to the absence of *Dee-D1* locus, and no conclusive remarks were made about the presence of 1DS.

Authors' Response:

Thank you for raising this point. We have considered it in the revised version at lines 273-276.

Comment:

Pg 17, Line 579, Mujeeb-Kazi, 2013 reference is missing.

Authors' Response:

We have corrected the suggested mistake.

In response to the comments by Reviewer 2

Thank you for your comments which had been incorporated in the revised manuscript.

Comment:

Though the *Dee-D1* loci is novel for endosperm development and its associated pleiotropic effect on spike morphology, it seems to be that this loci may be specific for Rye related crosses with hexaploid wheat. Is there any additional crosses made with the 1D deletion lines with hexaploid wheat lines and assessed the effect of this loci?

Authors' Response:

The Dt1DS line and five deletion lines along the long arm of chromosome 1D lack the *Dee-D1* locus. All these lines have low fertility and a significant decrease in MTS compared to CS. Crossing such lines with the euploid form of CS or other hexaploid wheat will allow obtaining F1 with one dose of chromosome 1D, but will not allow a reliable assessment of the degree of endosperm development in such plants, since the grains obtained on the F1 plant are already the second generation. In meiosis, during the formation of gametes in an F1 plant, a random divergence of the complete and deletion chromosomes 1D will occur, and the seeds from such a plant will be a mixture of different genotypes. Considering that the endosperm has a triploid number of chromosomes, this means that the development of endosperm in each specific flower will take place in the presence of either three full doses of chromosome 1D (3 doses of *Dee-D1*), 2 complete and one deletion chromosome (2 doses of *Dee-D1*), two deletion chromosomes and one complete (1 dose of *Dee-D1*) or three deletion chromosomes, i.e. without the *Dee-D1* locus. It is outwardly impossible to divide grains by genotype. Therefore, we considered it inappropriate to conduct such an experiment.

Comment:

Secondly the homeolog on chromosome 1B also showed some effect on endosperm development, eventually this should have been tested with few durum based crosses with 1D deletion lines. Nevertheless, this study identified promising loci for endosperm development in wheat. Apparently, there are more genes regulating the endosperm development in wheat as the volume of endosperm translates into grain yield in wheat and grain yield is governed by several QTL regions.

Authors' Response:

The presence of the *Dee-B1* homeolog on chromosome 1B was previously identified by us (Tikhenko et al., 2010) and was confirmed in this study by a comparative analysis of the productivity of the main spike, the ability to form viable grains in crosses with rye (Table 1, Table S2a) and the structure of the caryopsis using NMR (Fig. S6). However, the development of the endosperm both in lines and in hybrid caryopsis depends not only on the presence or absence of one or another *Dee* locus, but also on the number of copies of this locus in the endosperm cells, which is shown in Fig. S9. The conclusion that the *Dee-D1* locus has a stronger effect on endosperm development is based on the fact that two additional copies of *Dee-B1* in N1D/T1B line do not fully compensate for the absence of two copies of *Dee-D1* (Table 1, Table S2a; Fig. S6). Crossing of deletion lines of common wheat CS with tetraploid wheats will not lead to an increase in the number of *Dee-B1* loci in hybrid offspring, and therefore, when crossed with rye, such hybrids will produce non-viable caryopses as a result of the abortion of the hybrid endosperm, which is observed when crossing tetraploid wheats with rye. The production of wheat-rye hybrids from crosses of tetraploid wheat with rye is possible using an embryo rescue culture (Kruse 1974; Taira and Larter 1978; Raina 1984).

Comment:

Line 76: it should be 'world wheat' production

Authors' Response:

It had been changed in the revised version.

Comment:

Line 512 to 519, looks like there was a formula but it became like a box, correct this.

Authors' Response:

Thanks for highlighting this point, but it's a dash (-) and during the conversion from word to PDF became as a box. So, there is no formula included.

In response to the comments by Reviewer 3

Thank you very much for your critical review. All of your comments and corrections are incorporated in the revised manuscript.

Comment:

The authors have presented their case for the *Dee-D1* loci in wheat present on the short arm of hexaploid wheat's D1 chromosome. This is a novel discovery and the experiment were planned accordingly to make their case. It may be too early to claim this locus and its counterpart on B1 genome as one of the domestication genes. Authors have selected the lines carefully for making the crosses and creating either the addition and/or deletion lines. The phenotypes are contrasting, but they do show some phototropic effects compared to the reference CS line.

Authors' Response:

Thanks for your comments, we mapped *Dee-D1* on the long arm of hexaploid wheat's D1 chromosome that has a potential involvement of *Dee-D1* in the evolution not domestication. the pleiotropic effect of *Dee-D1* Our finding, showed that *Dee-D1* has a pleiotropic effect on spike morphology and productivity but we have to mention that this could be due to effect of other loci/genes located in the target area and involved in the regulation of spike morphology and fertility in hexaploid wheat.

Comment:

Considering that its a genetic loci the data looks sufficient. However as the wheat genome now has a much better quality reference genome, I would like to see which of the genes (sequenced loci) are missing in the region and do any of them or their orthologs from other species have seed/endosperm development and fertility phenotypes. Also check the EMBL-EBI gene expression atlas to see which of the sequenced loci/genes on D1S deleted part show preferred expression in the seed development process to narrow down their candidate genes. Provide all their data.

Authors' Response:

Thank you for raising such nice comment, In the CS wheat genome, the interval of deletion that includes the locus is located within the interval 259000000 bp to 494000000 bp at 1DL. A large number of genes are located within this interval (5655 genes) of which 2650 genes are high confidence (HC) genes that make it unattainable to suggest a candidate gene. The list of candidate genes within the physical deletion interval in the CS genome is available in **Tables S4**.

Comment:

Submit the GBS data to EMBL-EBI EVA. Authors have submitted the raw data to SRA, but that is not sufficient, they need to submit the SNP calls and GBS data to EVA along with the phenotype data. Need to describe a bit more about the parameters and depth of the sequencing etc.

Authors' Response:

We want to point out that we did not perform SNP calling because the cytogenetics stock analyzed in this study are in background Chinese Spring (i.e. the reference genotype used for genome assembly). The variants we are interested were not SNPs, but rather very large deletions. To discover them, we used a method based on read depth.

We have checked the EMBL-EBI EVA submission guidelines and to our understanding, data structure are geared towards SNP data. For this reason, we make the read depth data available under Digital Object Identifier (DOI) in the Plant Genomics & Phenomics Research Data Repository (PGP). The DOI is <http://dx.doi.org/10.5447/ipk/2020/25>.

We added a supplementary table (Supplementary Table 1) to report read counts and mapping statistics for the GBS samples.

Comment:

Reach out to GrainGenes wheat database to register their gene and share the sequencing, GBS and phenotype data for public use. Provide the parental and described lines to the local, national and international seed bank for others to use.

Authors' Response:

Thanks for the important suggestions. We have to mention that all plant materials (lines) were kindly provided by Dr. W. Jon Raupp from Wheat Genetic & Genomic Resources Centre at Kansas State University, Manhattan, USA. These lines are available from that source. We have contacted Taner Sen and Victoria Carollo-Blake of GrainGenes, who have been very helpful in setting up a website for Dee-D1, which will be accessible at:

[https://wheat.pw.usda.gov/cgi-bin/GG3/report.cgi?class=locus:name=DEFECTIVE+ENDOSPERM-D1+\(Dee-D1\)](https://wheat.pw.usda.gov/cgi-bin/GG3/report.cgi?class=locus:name=DEFECTIVE+ENDOSPERM-D1+(Dee-D1))

At the time of writing, the entire GrainGenes website is down due an IT issue. Therefore, we show a screenshot of the website:

Reference	Tikhenko-2020-submitted
Title	DEFECTIVE ENDOSPERM-D1 (Dee-D1) is crucial for endosperm development in hexaploid wheat
Year	2020
Remark	Submitted by Mascher et al. in 09.2020
Author	Tikhenko N [Show all 14]
Locus	DEFECTIVE ENDOSPERM-D1 (Dee-D1)

REVIEWERS' COMMENTS:

All my concerns were addressed. Recommend accepting for publication.